# Free-Form Surface Partitioning and Simulation Verification Based on Surface Curvature

**DOI:** 10.3390/mi13122163

**Published:** 2022-12-07

**Authors:** Hongwei Liu, Enzhong Zhang, Ruiyang Sun, Wenhui Gao, Zheng Fu

**Affiliations:** School of Mechatronical Engineering, Changchun University of Technology, Changchun 130012, China

**Keywords:** regional division, free-form surface, surface curvature, clustering algorithm, Voronoi diagrams

## Abstract

To address the problem of low overall machining efficiency of free-form surfaces and difficulty in ensuring machining quality, this paper proposes a MATLAB-based free-form surface division method. The surface division is divided into two stages: Partition area identification and area boundary determination. In the first stage, the free-form surface is roughly divided into convex, concave, and saddle regions according to the curvature of the surface, and then the regions are subdivided based on the fuzzy c-means clustering algorithm. In the second stage, according to the clustering results, the Voronoi diagram algorithm is used to finally determine the boundary of the surface patch. We used NURBS to describe free-form surfaces and edit a set of MATLAB programs to realize the division of surfaces. The proposed method can easily and quickly divide the surface area, and the simulation results show that the proposed method can shorten machining time by 36% compared with the traditional machining method. It is proved that the method is practical and can effectively improve the machining efficiency and quality of complex surfaces.

## 1. Introduction

With the rapid development of geometric modeling, more and more free-form surface models are available for the mathematical representation and design of complex industrial products [1]. Free-form surfaces are widely used in the design and manufacture of molds, automobiles, ships, aerospace, and commercial artwork [2], which have complex contours and high design and manufacturing accuracy requirements [3]. In order to ensure the machining quality of these large overlay molds, the surfaces are divided into several types of similar areas, different machining methods are designed, and the machining paths for each area are reasonably planned.

The production process of free-form surfaces is often divided into three stages: roughing, finishing, and benchwork [4]. In the roughing stage, material removal is the main focus, with high cutting forces and low requirements for surface quality. In the finishing stage, the material removal rate is low, the cutting force is low, and the surface quality is high. In the benchwork stage, the residual height is removed by grinding and polishing during the finishing process. Finishing and benchwork account for 78% of the total production time during the entire production process, which shows the importance of rational surface delineation.

Generally, free-form surfaces can be divided into convex, concave, saddle, and flat areas. The curvature between these surface pieces varies greatly and the value is uncertain. When grinding and polishing the surface using the grinding and polishing tool, the grinding force changes with the speed, thus reducing the grinding and polishing accuracy of the workpiece surface. Therefore, the surface area is reasonably divided according to the curvature of the surface, and the same grinding and polishing tool and process parameters are used for the same surface pieces, and the curvature of the surface in the same area changes within a certain range, so that the grinding force generated by the grinding and polishing tool during processing is more balanced and the grinding and polishing accuracy is improved.

A set of processing methods for dividing free-form surfaces is proposed, and its processing of surfaces can be divided into two processes: 1. partition region identification, and 2. region boundary definition. In the process of partition, area identification is further divided into surface coarse division and surface subdivision. The surface coarse division stage is based on the values of Gaussian curvature and mean curvature to classify them into convex, concave, and saddle surfaces, and then cluster analysis is performed using fuzzy c-means to get more accurate surface areas. The area boundaries are further defined by means of Voronoi diagrams.

## 2. Related Work

A lot of research has been done on the partitioning of specific free-form surfaces, Bendjebla et al. [5] proposed a new method for defining and classifying free-form surface machining features based on decomposition combination operations, where a free-form surface is first decomposed into a set of points and divided by a triangular mesh, and then combined into sub-surfaces with ten surface types based on shape descriptors. Shi Qun et al., [6] based on the triangular mesh model of a free-form surface, used a directed Voronoi diagram algorithm to divide the free-form surface, and realized the complex surface area division of cutter axis motion optimization. Liu et al. [7] used T-spline to divide the free-form surface into convex, concave, and flatter areas, and then introduced the sliding frame method to describe the boundaries of the areas. You et al. [8] used curvature and normal vector weight fuzzy clustering algorithms to automatically subdivide a monolithic surface into multiple surface slices. Bey et al. [9] approximated the surface by constraining the minimum number of triangles to identify the local shape of each vertex based on curvature, classifying it as concave, convex, planar, concave-spreadable, convex-spreadable, and saddle-shaped. Van et al. [10] developed a MATLAB program to perform surface partitioning, which first partitions the surface into planar, convex, concave, and saddle regions based on surface curvature, and then define the surface boundaries by the Freeman chain code technique.

Nguyen et al. [2], also based on curvature and Freeman chain code techniques to delineate surface areas, introduced some advanced CAD/CAM techniques for free-form machining modeling and toolpath planning in [11] to apply this method of surface delineation to five-axis machining. Tuyen [12] et al. used the same method to partition the surface and applied it to 3-axis machining based on the partition results. They ensure the quality of surface machining while reducing the machining time by optimizing the tool and tool path when machining free-form surfaces. However, they all use B splines to describe the free-form surfaces, which result in simpler free-form surfaces with little surface variation, and are also limited by B spline input and cannot handle NURBS surfaces.

Roman [13] used the fuzzy c-means method to subdivide the free-form surfaces, and the boundary of each surface patch was delineated in the u-v plane using the nearest neighbor method. Moreover, in [14], the surfaces were subdivided into patches using the k-means clustering method and the boundaries between patches were identified by the minimum intra-class distance method. This method causes the number of patches generated to exceed the number of concave, convex, and saddle areas of the surface, which will increase the time for subsequent machining. Herraz et al. [15] introduced various clustering algorithms for free-form surface segmentation by region processing and analyzed and compared representative algorithms in three major classes of unsupervised algorithms: k-means algorithm, RPCL algorithm, and HAC algorithm. The authors concluded that using the k-means algorithm with Euclidean metrics and dedicated feature vectors would be a better choice. However, he only made a longitudinal comparison and did not further make a cross-sectional comparison between k−means, k−medoids, and fuzzy c-means algorithms.

Elber and Cohen [16] used a mixture of symbolic and numerical methods to calculate curvature, using scalar and vector fields for the analysis of surfaces, respectively, ternating the surfaces into saddle, concave and convex regions and delimiting the surface curvature. He et al. [17] divided the surface into different regional facets with curvature-induced vector fields and analytically processed the scalar and vector fields and contours, covering each segmented patch with a regular quadrilateral of constant length.

After determining the distribution of each face piece on the surface, the definition of the boundary between each face piece has a critical impact on the subsequent machining. Liu et al. [18] proposed a tool trajectory method for machining on a five-axis machine. After dividing the free-form surface into regions using the k-means clustering algorithm, the boundaries between the regions were defined using B spline curves. In [19], Mao, based on the K−d−tree K−nearest neighbor (KNN) algorithm, formed a number of point clusters after searching the discrete points of the surface, and the boundary line of the surface slice was obtained by linearly connecting the outermost points of each discrete point cluster. After dividing the surface into different regions, Xie et al. [20] proposed a concept of “easy grinding zones” and introduced an algorithm to automatically select a compatible belt grinding mode for each zone. Yin [21] analyzed the variation law of machining time and surface roughness on spindle speed, feed rate, cutting width, depth of cut, and surface machining complexity after completing the zone division, and used the weighted summation method to determine the optimal grinding parameters for different machining complexities. Zhang et al. [22] proposed a new damping cloth (DC) tool for non-contact polishing and a chemically enhanced non-Newtonian ultrafine (CNNU) polishing solution to achieve sub-nanometer roughness on aspheric optical molds.

In summary, using NURBS instead of B splines to describe free-form surfaces when calculating surface curvature can handle a wider variety of more complex surfaces. In the subdivision of the surface, k-means and fuzzy c-means algorithms are comprehensively compared, and the fuzzy c-means algorithm with a better clustering effect is selected. In the final determination of the surface slice boundaries, Voronoi diagrams are used to determine the boundaries of each region, and the program runs stably and quickly, while the division results are reliable.

## 3. Free-Form Surface Representation Geometry

### 3.1. Free-Form Surface Representation

Generally, free-form surfaces can be designed directly on a computer using control points, such as Bezier surfaces, B-spline surfaces, and non-uniform rational B-spline surfaces (NURBS) [23]. These surfaces are a common way to describe free-form surfaces. NURBS surface modeling not only has the advantages of each of these surface models, but it can also adjust the shape of the surface by modifying the control vertices and weight factors of the curved surface, so that the constructed free-form surface is closer to the true value; in this paper, NURBS is used to define free-form surfaces. A NURBS surface is usually represented by a rational fraction, rational basis function, and homogeneous coordinates. No matter what method is adopted, the curvature calculation and region division of the surface are the same. Therefore, this paper chooses rational fraction to describe the surface, and its NURBS description form is as follows:(1)$$S:P(u,v)=\sum_{i=1}^{k}P_{i}(u,v)=\sum_{i=1}^{k}[u^{3},u^{2},u,1]Q_{i}{\left[v^{3},v^{2},v,1\right]}^{T}\;$$
where P(u,v) is the parametric equation of the free-form surface S;$P_{i}(u,v)$ is the parametric equation of the ith surface slice; $Q_{i}$ is a 4 × 4 square matrix indicating the vector equation of the ith surface slice; u,v is the two parameters of the parametric surface equation and satisfies 0≤u,v≤1; k is the number of surface slices. The free-form surface selected in this paper is shown in Figure 1.

When performing surface discretization, using too many parameter lines results in an excessive number of parameter points. This has little effect on the surface division results, but instead increases the workload of the algorithm and reduces the processing speed. If too few lines are used, the number of discretization points will not be large enough, and the surface division results will be inaccurate. Therefore, the selection of parametric lines should be combined with the size of parts, the complexity of surfaces, and the computing power of computers. In summary, this paper uses 25 × 25 parametric lines to discretize free-form surfaces.

### 3.2. Surface GEOMETRY

Given a free-form surface $S(u,v)=[S_{x}(u,v),S_{y}(u,v),S_{z}(u,v)]$. Here are some geometric parameters of the surface.

(1) Normal vector of a point on the surface

The unit normal vector of the surface S(u,v) at the point (u,v) can be calculated from the following equation:(2)$$S(u,v)=\frac{\left(S_{u}\times S_{v}\right)}{\left|S_{u}\times S_{v}\right|}$$
where $S_{u}$ and $S_{v}$ are the tangential vectors along the direction of the u and v parameters.

(2) Mean curvature (*H*) and Gaussian curvature (*K*)

The average curvature H of the surface S(u,v) at the point P(x,y,z) is expressed as:(3)$$H=\frac{1}{2}(\frac{EN-2FN+GL}{EG-F^{2}})=\frac{1}{2}(K_{\max}+K_{\min})$$

The Gaussian curvature *K* is expressed as:(4)$$K=\frac{LN-M^{2}}{EG-N^{2}}=K_{\max}K_{\min}$$
where *E*, *F*, *G*, *L*, *M,* and *N* are the parameters of the first and second fundamental forms:(5)$$E=\frac{\partial S}{\partial u}\frac{\partial S}{\partial u};F=\frac{\partial S}{\partial u}\frac{\partial S}{\partial v};G=\frac{\partial S}{\partial v}\frac{\partial S}{\partial v}$$
(6)$$L=n\frac{\partial^{2}S}{\partial u^{2}};M=n\frac{\partial^{2}S}{\partial u\partial u};N=n\frac{\partial^{2}S}{\partial v^{2}}$$

$K_{\max}$ and $K_{\min}$ are the principal curvatures, which are the maximum and minimum values of the normal curvature and can be given by:(7)$$K_{\max}=H+\sqrt{H^{2}-K}$$
(8)$$K_{\min}=H-\sqrt{H^{2}-K}$$

Also Gaussian curvature and mean curvature need to satisfy:(9)$$H^{2}\geq K$$

The two principal curvatures $K_{max}$, $K_{min}$, Gaussian curvature *K*, and average curvature *H* of the surface are all valid for the description of the curvature. The principal curvatures of the surface are direction independent and are the extrinsic properties of the surface. Numerically, the main curvature is more sensitive to noise than the average curvature *H* but lower than the Gaussian curvature *K*. At the same time, the computational workload of the principal curvature is also higher than that of the Gaussian curvature *K* and the average curvature *H*. The principal curvature must also be used in pairs due to its directivity. The Gaussian curvature *K* and the average curvature *H* have nothing to do with the direction. The Gaussian curvature *K* is the internal characteristic of the surface, and the average curvature is the external characteristic of the surface. The average curvature *H* and the boundary curve together determine the unique surface. The Gaussian curvature function can determine the only convex surface. Therefore, this paper uses Gaussian curvature *K* and average curvature *H* to preliminarily determine the concave, convex, and saddle regions of free-form surfaces.

The sign of Gaussian curvature *K* and mean curvature *H* at a point on the surface can determine the type of the point and the local surface shape; summarized as shown in Table 1.

## 4. Division of Free-Form Surfaces

### 4.1. Surface Rough Division

The purpose of dividing the surface area is to divide the freeform surface into several areas with the same characteristics, and the same surface piece is processed with the same grinding and polishing tools and process parameters, which can effectively improve the efficiency of processing the freeform surface and the surface finish. In this paper, the surface is divided into a convex region (flat region), concave region, and saddle region based on Gaussian curvature K and mean curvature H. Since there is no overcutting during convex and flat surface machining and the tool size requirement is low, the convex and flat surface area is classified as part of the process. This reduces the number of surface slices divided and reduces the calculation time while improving processing efficiency. Figure 2 shows the flow diagram for the coarse partitioning of surfaces.

The above algorithm is implemented by MATLAB. After determining the positive and negative values of Gaussian curvature and mean curvature, the grid points on the surface are initially coarsely divided into groups and the convex, concave, and saddle areas of the surface are indicated by □, ◇, and ▽ respectively.

### 4.2. Free-Form Surface Subdivision

After the above processing, the free-form surface has been coarsely divided into three types of area slices: convex (planar), concave, and saddle piece surfaces. Then the surface is further divided by the fuzzy c-means clustering method FCM (Fuzzy c-means).

The idea of clustering is to maximize the similarity between objects that are divided into the same cluster and minimize the similarity between different clusters. The conventional K−means algorithm divides the data rigidly and divides it according to rigid standards, and the result of the division is either one or the other. The fuzzy c-means algorithm is a flexible fuzzy division. Its idea is to maximize the similarity between objects that are divided into the same cluster, while the similarity between different clusters is the smallest. The membership degree in the k−means algorithm has only two options: 0 or 1, and the membership degree of the fuzzy c-means is any number in the [0 1] interval, which makes the fuzzy c-means clustering effect better than the k−means algorithm; Figure 3 below shows the processing effect of the two algorithms on the Bunny point cloud data in the Stanford University point cloud library. It can be clearly seen that the fuzzy c-means has a more reasonable clustering effect.

The content of the fuzzy clustering method is as follows: let the sample space be $A=\{x_{1},x_{2},\ldots ,x_{n}\}$, where each element contains s attributes, fuzzy clustering is to divide x into c(2≤c≤n) classes, c are clustering centers. The objective function of FCM is as follows:(10)$$J(U,V)=\sum_{i=1}^{c}\sum_{j=1}^{n}u_{ij}^{m}||v_{i}-x_{i}|{|}^{2}=\sum_{i=1}^{c}\sum_{j=1}^{n}u_{ij}^{m}d_{ij}^{2}$$
where $u_{ij}$ is the affiliation of the jth element of the sample space A to the ith class center, m is the fuzzy index. m∈[1,∞), the larger the value of m, the higher the fuzziness of the classification, usually the value of m is 2; $d_{ij}=||v_{i}-x_{i}||$ is the Euclidean distance between the ith cluster center and the jth data point; $V=\{v_{ij}\}$ is the cluster center, $U=\{u_{ij}\}$ is the affiliation matrix; the affiliation matrix U takes values between 0, 1, and the sum of the affiliation of a data is equal to 1:(11)$$\sum_{i=1}^{c}u_{ij}=1,\forall j=1,\ldots ,n$$

Equations (12) and (13) are used to calculate the cluster center $v_{ij}$ and the membership degree $u_{ij}$, respectively:(12)$$v_{ij}=\sum_{j=1}^{n}u_{ij}^{m}x_{i}/\sum_{j=1}^{n}u_{ij}^{m}$$
(13)$$u_{ij=}{1/\sum_{k=1}^{c}\left(\frac{d_{ij}}{d_{kj}}\right)}^{2/\left(m-1\right)}$$

The fuzzy C-mean clustering algorithm flows as follows:

Step 1: Initialize the membership matrix *U* with a random number with a value between (0, 1), so that it satisfies the constraints in Equation (11);

Step 2: Use formula (12) to calculate the c cluster centers $v_{ij}$;

Step 3: Calculate the value function according to the formula (10). If it is less than a certain threshold, or its change from the last value function value is less than a certain threshold, the algorithm stops;

Step 4: Calculate the new *U* matrix with (13). Return to step 2.

### 4.3. Surface Boundary Definition

After the above process, the free-form surface has been divided into three types of regions, but the boundaries of these regions have not been completely defined. In order to determine clear boundaries, Voronoi diagrams are used to define the boundaries in this paper. Voronoi diagrams are also known as Thiessen polygons or Dirichlet diagrams, which were proposed by Dirichlet in 1985. The Voronoi diagram is a division of the space plane, which is characterized by the fact that any position within a polygon is the closest to the site of the polygon, and far from the site in the adjacent polygon, and each polygon contains and only contains one site (Figure 4). If the Voronoi diagram is established at the center of the area block, the established Voronoi area coincides with the area block, and the Voronoi diagram becomes the boundary of the surface slice.

Given a plurality of stations, each station is located in the Voronoi region included in its boundary, and all Voronoi region boundaries form the Voronoi diagram. The Voronoi diagram can be built in two steps: 1. construction of the Delaunay triangular grid; 2. connecting the outer circle of the triangular grid to a line. We demonstrate these two processes with 23 random points, and the running results are shown in Figure 5 and Figure 6 below.

Definition of Delaunay Triangle Mesh: It is composed of a series of connected but non-overlapping triangles. It has two properties, 1. Empty circle property: the Delaunay triangle network is unique, any four points in space cannot be co-circular, while the external circle of these triangles does not contain any other points of this surface domain; 2. Maximizing the minimum angle characteristic: The minimum angle of the triangle formed by the Delaunay triangle is the largest, specifically, the diagonal of the convex quadrilateral is formed in two adjacent triangles. After a mutual exchange, the minimum angle of the inner angle of the convex quadrilateral will not increase, as shown in Figure 4.

In this paper, an improved point-by-point insertion method, the Bowyer-Watson algorithm, is used to generate Delaunay triangulation. Its main steps are as follows:

1. Construct a super triangle containing all scatter points;

2. Insert each point in turn, the new point inserted in the grid, draw external circles in the triangle associated with it, record all the triangles whose external circles contain the new point, delete the common edges affecting the triangle and replan the new edges;

3. Recording the triangles into the grid;

4. Repeat step 2 until all scattered points are inserted;

5. Delete the super triangle and its related triangles.

Each vertex of the Voronoi diagram is the center of the outer circle of the Delaunay triangular mesh, so the second step of the algorithm is much simpler:

Find the centers of circumscribed circles of all triangles;

Connect the centers of the circles in order according to the relationship between the triangles.

Voronoi diagrams are applied to two-dimensional parametric surface space. The free-form surface used in this experiment is a three-dimensional data structure, so it is necessary to map the projection of the three-dimensional clustering centers to the two-dimensional parametric surface space, construct Voronoi diagrams with these two-dimensional clustering centers as sites, and generate surface boundaries in two-dimensional space to show the results of free-form surface region partitioning.

## 5. Program Flow

The experiment in this paper uses MATLAB R2020b, the CPU is AMD Ryzen 5 4600H with Radeon Graphics, the graphics card is NVIDIA GeForce GTX 1650 4 GB, and the memory is a 16 G computer for verification.

After calculating and analyzing the Gaussian curvature (K) and mean curvature (H) of the free-form surface, the surface is divided into three major region types and represented by special symbols. This is shown in Figure 7.

A fuzzy c-means clustering method was used to optimize the convex regions based on the number of clusters and cluster centers while optimizing the saddle-shaped regions in a similar way. In Figure 8 below, different notations are used to mark different regions, and a total of 11 blocks of regions are shown, including four groups of convex regions (planar regions), three groups of concave regions, and four groups of saddle regions, while the clustering centers are positioned with (☆). Here, due to the large area of the convex and saddle areas at the beginning, the larger areas are divided again for the convenience of subsequent grinding and polishing, and to adapt to the grinding and polishing tool head.

Using the optimized clustering centers as sites, Voronoi diagrams are generated to define the region boundaries, as shown in Figure 9. After this process is processed, the free-form surface partition has been completed and each surface slice has been fully defined.

## 6. Simulation Experiment

After the above-mentioned program processing, the surface area division has been completed. However, because the Voronoi diagram division result is a two-dimensional parameter space, it is necessary to convert the division result into a three-dimensional parameter space. Here, we use the projection command function of the generative shape design in Catia software to project the divided area and boundary onto the free-form surface model to be machined, the material is defined as 45 steel, as shown in Figure 10.

In order to verify the rationality of surface subdivision, three-axis NC milling in the free-form surface machining module of CatiaV5R21 is used for simulation machining. At the same time, according to the previous curvature calculation, the minimum radius of the concave and saddle area is calculated as 1.4753 mm and 3.3675 mm. In order to avoid overcutting while not wasting machining time, 2 mm and 6 mm diameter tools are used for milling in the concave area and saddle area respectively, and the specific machining scheme is as follows.

Step 1: Roughing of two surface models, noted as Sample 1 and Sample 2, using a ball-ended milling cutter with diameter D1 = 10 mm, feed rate = 600 mm/min, and a scallop height of 0.5 mm, as shown in Figure 11a;

Step 2: Overall finishing of sample 1 using a ball-head milling cutter with diameter D2 = 2 mm, feed rate = 1000 mm/min, and a scallop height of 0.05 mm, as shown in Figure 11b;

Step 3: Sample 2 was machined using the same machining parameters except for the tool size, which was divided into three processes: (1) milling the convex area with a ball-head milling cutter of diameter D3 = 8 mm, as shown in the blue area of Figure 12b; (2) milling the saddle area with a ball-head milling cutter of diameter D4 = 6 mm, as shown in the yellow area of Figure 12c; (3) milling the concave area with a ball-head milling cutter of diameter D2 = 2 mm, as shown in the purple area in Figure 12d.

In order to compare the processing time of finishing between domain processing and traditional processing methods, some restrictions should be added to both machining methods when generating the toolpaths, and the following machining parameters should be the same regardless of the tool used for machining: spindle speed (7000 rev/min), feed rate (1000 mm/min), depth of cut (0.5 mm), residual for roughing (0.5 mm), scallop height (0.05 mm). Moreover, to avoid overcutting in the concave area, the tool used in sample 2 to machine the concave area was used in the conventional machining method. The process of sample 1 is shown in Figure 11 and the process of sample 2 is shown in Figure 12, and the tool path length and machining time are shown in Table 2.

With the above machining parameters and tool path strategy, the tool path is significantly shortened compared to conventional machining methods, saving machining time.

## 7. Conclusions

A MATLAB program was designed and developed to perform the surface division work. By inputting the surface parameter equation or the point cloud data obtained by scanning the surface, the area of the free-form surface can be divided, and the number of surface pieces generated by division can be limited and adjusted according to the actual machining requirements and production conditions. This avoids the problem of too long machining time or over-cutting of the cutter caused by too many or too few surface pieces and effectively improves the efficiency and quality of free-form surface machining.

1. Define a free-form surface model with NURBS, calculate the curvature of the free-form surface, and use the mean curvature and Gaussian curvature to initially classify the surface into a concave region, convex region, and saddle region by analyzing and comparing the effects of principal curvature with mean curvature and Gaussian curvature on the surface properties. By comparing the clustering effects of fuzzy c-means and k−means algorithms, it was decided to use the fuzzy c-means algorithm for subdividing the surface pieces. The boundary between the surface slices is determined by the Voronoi diagram in the MATLAB library based on the clustering centers generated by the clustering, and the division of the surface processing area is completed.

2. The method was verified using the simulation machining module in Catia software. The simulation results show that compared with the traditional machining method, the method in this paper can ensure the machining quality while saving 36% of the machining time, which can effectively improve the machining efficiency.

In the next step, the simulation results will be verified experimentally, and the free-form surface grinding and polishing study will be carried out to improve the smoothness between the surface pieces through the NURBS surface stitching method and the corresponding processing method, and to optimize the process parameters and rationalize the grinding and polishing path to further improve the surface quality.

## Figures and Tables

**Figure 1 micromachines-13-02163-f001:**
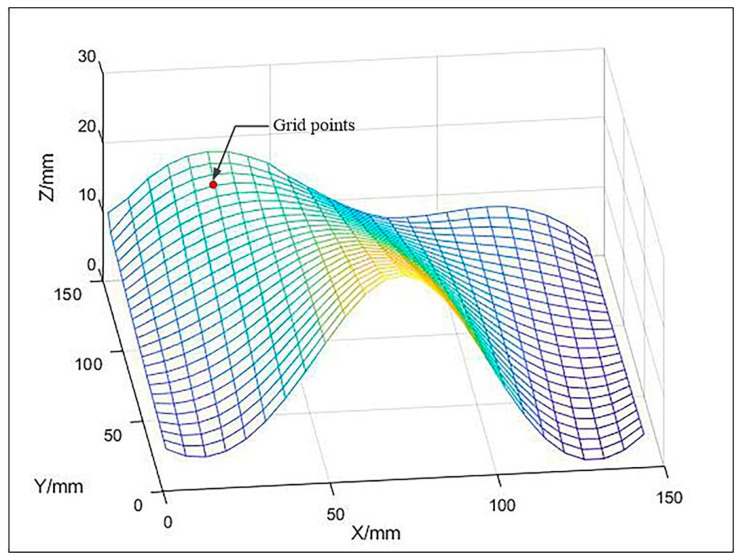
Mesh Surface and Grid points.

**Figure 2 micromachines-13-02163-f002:**
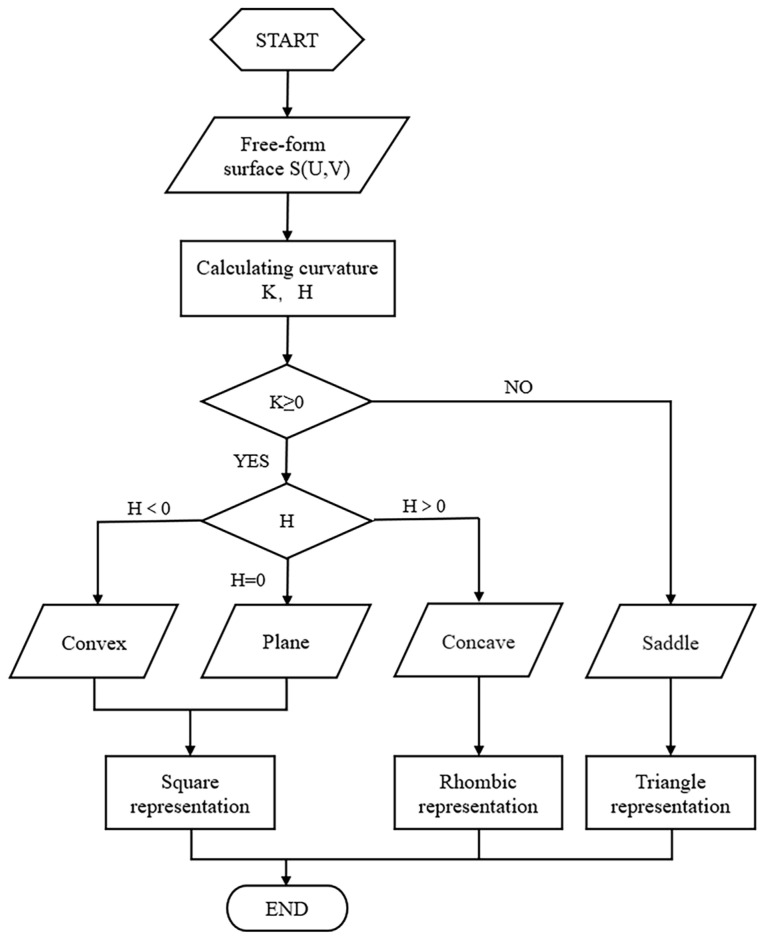
Flow chart of a rough surface.

**Figure 3 micromachines-13-02163-f003:**
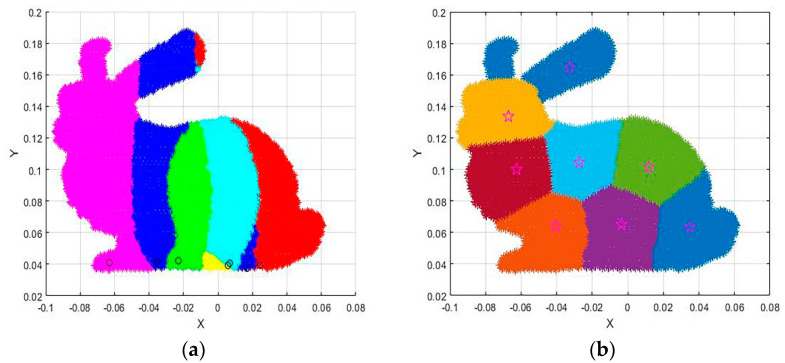
Comparison of FCM and K−means clustering effects under the same number of clusters: (**a**). K−means Algorithm, the circles represent the clustering centers of K−means; (**b**). FCM Algorithm, the stars represent the clustering centers of FCM.

**Figure 4 micromachines-13-02163-f004:**
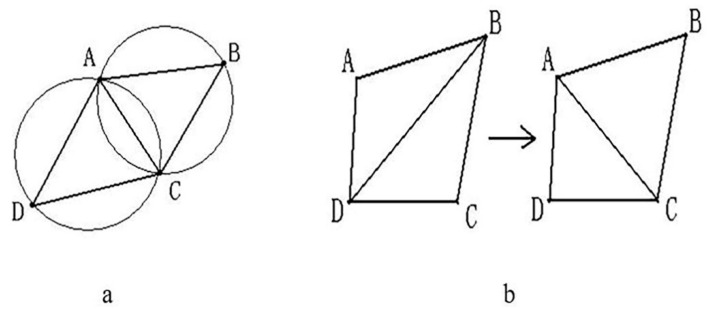
(**a**) Empty characteristic, (**b**) Minimum angle maximization characteristic. A, B, C, D are any four points in space.

**Figure 5 micromachines-13-02163-f005:**
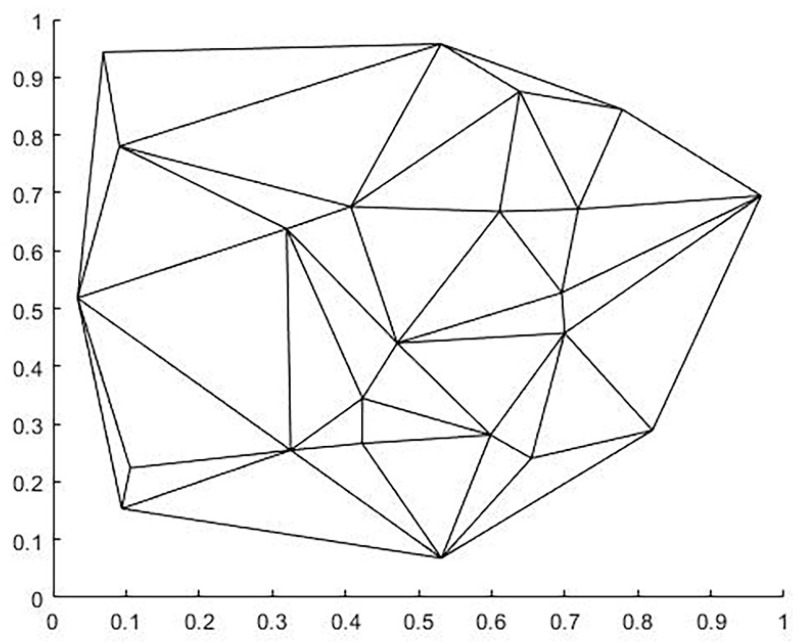
Delaunay triangulation.

**Figure 6 micromachines-13-02163-f006:**
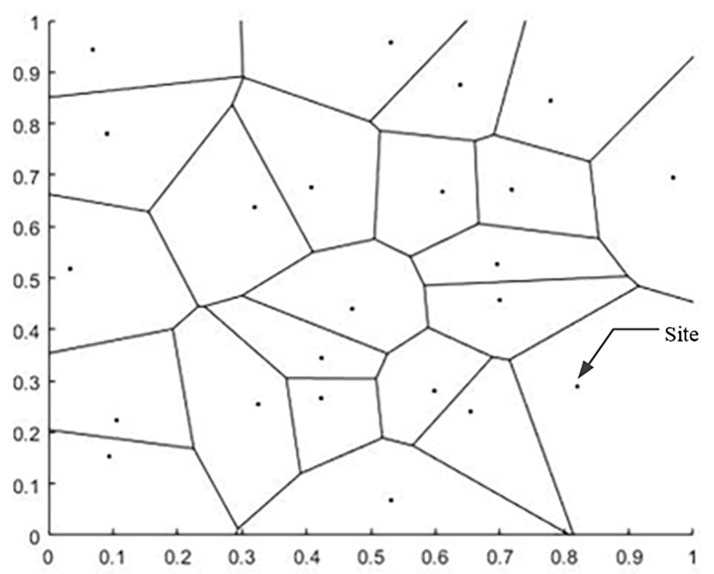
Voronoi diagram of 23 sites and regions.

**Figure 7 micromachines-13-02163-f007:**
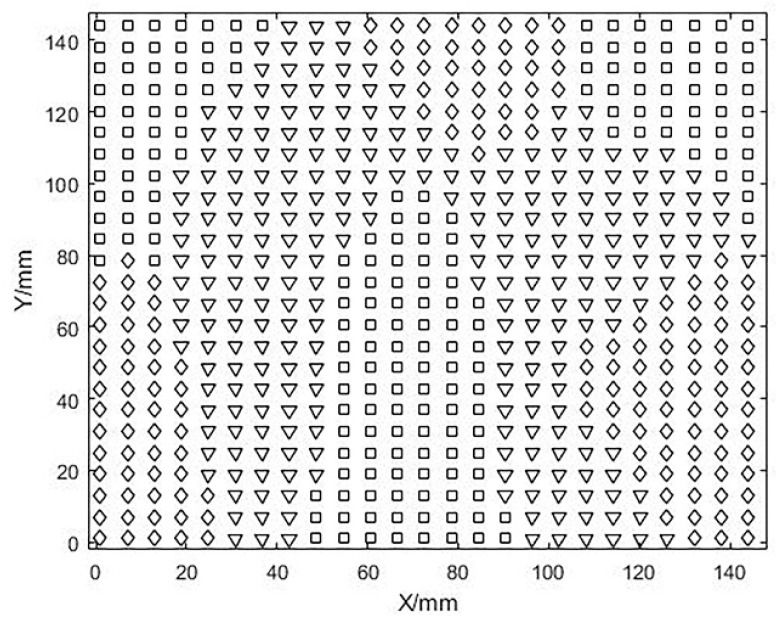
Preliminary division of the surface based on curvature, □ stands for convex area, ▽ stands for saddle area, ◇ stands for concave area.

**Figure 8 micromachines-13-02163-f008:**
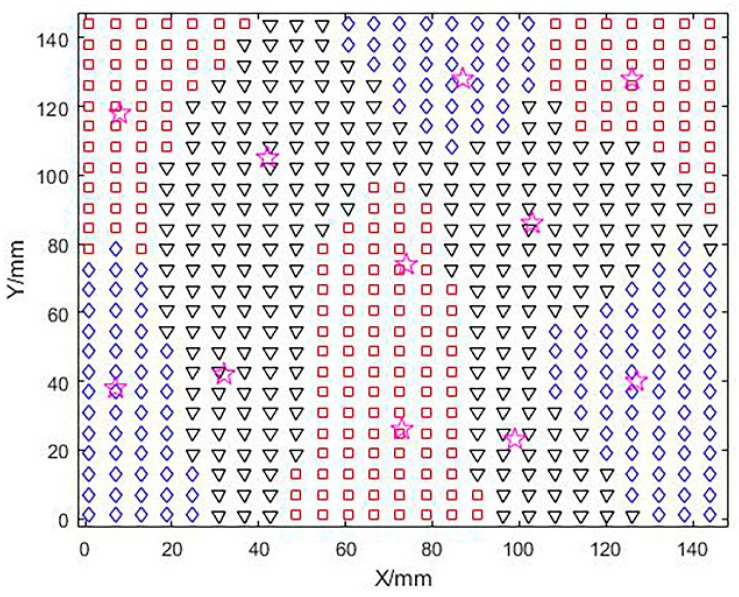
Cluster center and subdivision area.

**Figure 9 micromachines-13-02163-f009:**
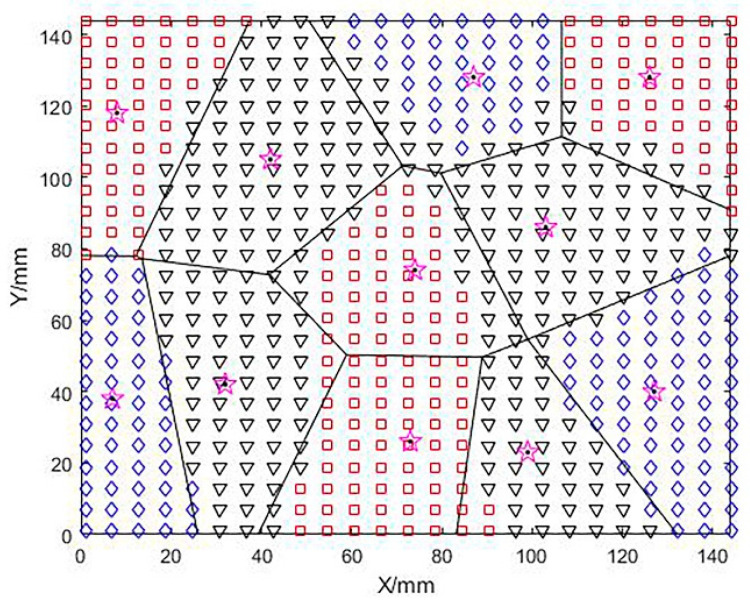
The result of boundary division of Voronoi diagram.

**Figure 10 micromachines-13-02163-f010:**
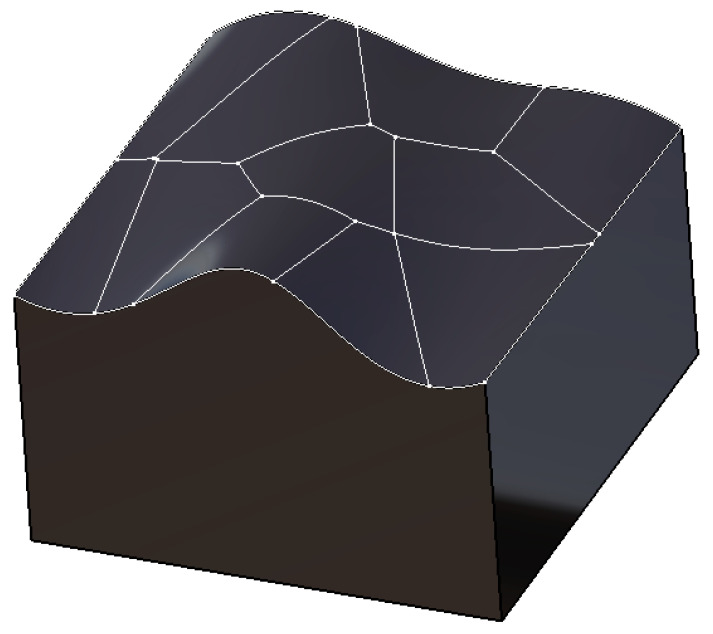
3D projection surface segmentation.

**Figure 11 micromachines-13-02163-f011:**
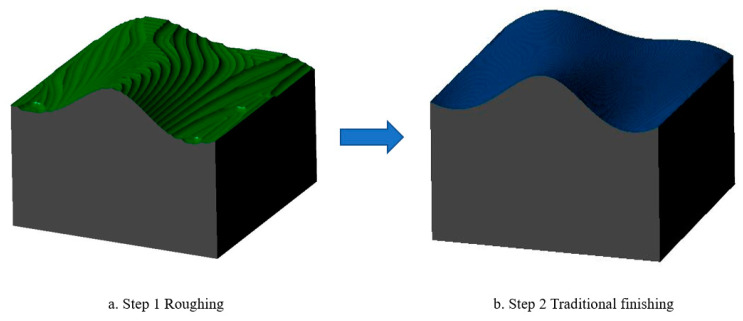
Sample 1 simulation processing process, (**a**) Roughing, (**b**) Traditional finishing.

**Figure 12 micromachines-13-02163-f012:**
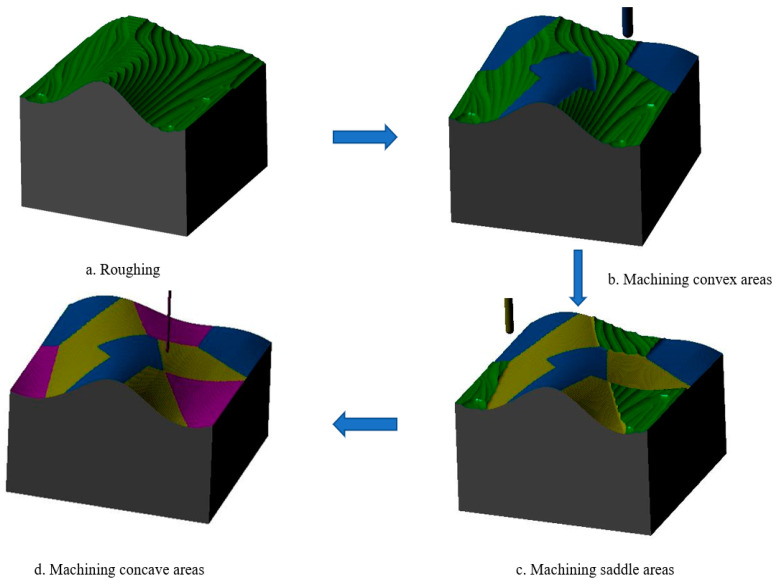
Sample 2 sub-domain machining process: (**a**) rough machining process; (**b**) convex area machining process; (**c**) saddle area machining process; (**d**) concave area machining process.

**Table 1 micromachines-13-02163-t001:** Relationship between surface shape and geometric parameters.

Gaussian Curvature (K)	Mean Curvature (H)	Point Feature	Local Surface Shape
K > 0 K > 0 K < 0 K = 0 K = 0 K = 0	H > 0 H < 0 H > 0 or H < 0 H = 0 H > 0 H < 0	Oval Oval Hyperbola Parabola Parabola Parabola	Concave area Convex area Saddle area Flat area Concave area Convex area

**Table 2 micromachines-13-02163-t002:** Comparison of simulation results.

	Simulation Results	Toolpath Length/(mm)	Processing Time/(s)
Processing Method			
Traditional method		45,280	2717
Proposed method		29,110	1747

## Data Availability

Not applicable.

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
