# Peer review of "Free-Form Surface Partitioning and Simulation Verification Based on Surface Curvature"

_micromachines, 2022, doi:10.3390/mi13122163_

Round 1

Reviewer 1 Report

The development of CAD/CAM systems and machine tools is leading to the possibility of machining parts with increasingly complex shapes. One area of work aimed at increasing the productivity and efficiency of manufacturing processes is the machining of free-form surfaces.  The article presented here, therefore, addresses a topical and important topic.  The article is well-written and easy to read. For the most part - apart from a few fragments - it is easy to follow the authors' reasoning. In a few places, however, the text could use some clarification:

1.       In section 4.1, an algorithm for classifying surfaces due to the parameters H and K are presented. What is the relationship between the 'grid point' and the surface S(U, V)? Is this point the midpoint of this surface or one of the corners? It might be useful to illustrate this in the figure, e.g. mark one such surface and grid point in figure 1.

2.       In point 6 simulation studies are described. Is Step 1 - roughing,  the same for traditional and domain processing methods?  Does Sample 1 refer to the conventional method and Sample 2 to the domain processing method?

3.       In the summary, the authors used the phrase "a set of contrast experiments are designed" - while only one experiment was conducted.

In addition, please refer to the following comments:

4. On what basis was the mesh size selected during preliminary surface division? Was the mesh size dictated by any specific indications? The mesh size will probably have an impact on the classification of the surface sections. How might this affect the final division and machining process?

5. Besides machining time, surface quality is a key parameter. Have the final machining results been compared in any way?

6. What advantages do the authors see in the proposed approach for machining complex surfaces compared to existing solutions? What are the limitations and main areas of application of the presented approach - with which types of surfaces can it be more effective than others, and with which not.

Author Response

Dear Professor,

Thank you very much for your comments and suggestions.

Those comments are all valuable and very helpful for revising and improving our paper, as well as the important guiding significance to our researches. We have studied comments carefully and have made correction which we hope meet with approval. Revised portion are marked in blue in the paper. The main corrections in the paper and the responds to the reviewer’s comments are as flowing:

  1. Comment: In section 4.1, an algorithm for classifying surfaces due to the parameters H and K are presented. What is the relationship between the 'grid point' and the surface S(U, V)? Is this point the midpoint of this surface or one of the corners? It might be useful to illustrate this in the figure, e.g. mark one such surface and grid point in figure 1.

Response: The process of surface discretization is to cover the entire surface with the parametric lines of the surface (U and V lines), and the intersection of the U and V lines is the surface grid point to be calculated. The picture has been modified in the text.

  1. Comment: In point 6 simulation studies are described. Is Step 1 - roughing,  the same for traditional and domain processing methods?  Does Sample 1 refer to the conventional method and Sample 2 to the domain processing method?

Response: The same processing method was used for both samples in the roughing stage. Sample 1 is the conventional method and sample 2 is the sub-domain processing method. The picture has been modified in the text.

  1. Comment: In the summary, the authors used the phrase "a set of contrast experiments are designed" - while only one experiment was conducted.

Response: The text has been modified.

  1. Comment: On what basis was the mesh size selected during preliminary surface division? Was the mesh size dictated by any specific indications? The mesh size will probably have an impact on the classification of the surface sections. How might this affect the final division and machining process?

Response: When performing surface discretization, using too many parameter lines results in an excessive number of parameter points, which has little effect on the surface division results, but instead increases the workload of the algorithm and reduces the processing speed. If too few lines are used, the number of discretization points will not be large enough, and the surface division results will be inaccurate. Therefore, the selection of parametric lines should be combined with the size of parts, the complexity of surfaces and the computing power of computers. In summary, this paper uses 25×25 parametric lines to discretize free-form surfaces.

  1. Comment: Besides machining time, surface quality is a key parameter. Have the final machining results been compared in any way?

Response: Free-form surface production is generally divided into three stages: rough milling, fine milling and polishing. Because of the influence of COVID-19 epidemic, it is impossible to carry out experimental verification now. After the epidemic situation in COVID-19 has improved, we will verify the simulation results by experiments, carry out research on grinding and polishing of curved surfaces, optimize process parameters and rationally plan grinding and polishing paths to further improve the surface quality of curved surfaces.

  1. Comment: What advantages do the authors see in the proposed approach for machining complex surfaces compared to existing solutions? What are the limitations and main areas of application of the presented approach - with which types of surfaces can it be more effective than others, and with which not.

Response: This method not only improves the efficiency of surface machining compared with traditional machining methods, but also prevents tool overcutting and effectively ensures the quality of surface machining. Secondly, the algorithm is general enough to handle a wide range of spline defined surfaces, and can also be applied in reverse engineering to deal with free-form surfaces fitted by point clouds. The limitation is that only single-surface surfaces can be handled, and further research is needed for models with multiple surfaces.

Reviewer 2 Report

- for what kind of raw material (aluminium, steel etc) are cutting conditions (feed, speed, depth of cut) applied ?

- what milling strategy was applied for roughing as well as for finishing operations?

- please correct units for spindle speed (rev/min not r/min)

- suggestions for future research should be summarized (like real sample production and comparison with simulated one)

-  what kind of machine tools was selected in CATIA for simulation (3 or 5 axis CNC)

- what about boundaries between areas machined with different strategy (residual material, different surface quality etc.). Please give an opinion on this issue within conclusions. This is very important issue to be considered when machining high quality free-form surfaces.

- Fig. 11 and Fig. 12 not very clear, bit of confusing. In Fig. 11 is step 1 - roughing and step 2 is also traditional roughing (it should be probably written - "traditional finishig")? On the other hand, in Fig. 12 is simulated workpiece after roughing operation missing. 

Author Response

Dear Professor,

Thank you very much for your comments and suggestions.

Those comments are all valuable and very helpful for revising and improving our paper, as well as the important guiding significance to our researches. We have studied comments carefully and have made correction which we hope meet with approval. Revised portion are marked in blue in the paper. The main corrections in the paper and the responds to the reviewer’s comments are as flowing:

  1. Comment: for what kind of raw material (aluminium, steel etc) are cutting conditions (feed, speed, depth of cut) applied ?

Response: Model material is common metal 45 steel, the original Figure 10 has been modified, as shown in Figure 10 below. Cutting conditions: spindle speed (7000 rev/min), feed rate (1000 mm/min), depth of cut (0.5 mm), residual for roughing (0.5 mm), scallop height (0.05 mm).

  1. Comment: what milling strategy was applied for roughing as well as for finishing operations?

Response: In rough milling, rough milling with equal height and lower layer is adopted, while in finish milling, guided cutting is adopted.

  1. Comment: please correct units for spindle speed (rev/min not r/min)

Response: Modified in the text.

  1. Comment: suggestions for future research should be summarized (like real sample production and comparison with simulated one)

Response: Revisions have been made in the text based on the suggestions.

  1. Comment: what kind of machine tools was selected in CATIA for simulation (3 or 5 axis CNC)

Response: 3 axis CNC.

  1. Comment: what about boundaries between areas machined with different strategy (residual material, different surface quality etc.). Please give an opinion on this issue within conclusions. This is very important issue to be considered when machining high quality free-form surfaces.

Response: Additional modifications have been made in the conclusion. The size of the boundary processing domain between adjacent segmented surfaces is determined through the research of stitching algorithm, and the corresponding processing method is used for processing.

  1. Comment: Fig. 11 and Fig. 12 not very clear, bit of confusing. In Fig. 11 is step 1 - roughing and step 2 is also traditional roughing (it should be probably written - "traditional finishig")? On the other hand, in Fig. 12 is simulated workpiece after roughing operation missing. 

Response: The picture has been modified in the text.

Reviewer 3 Report

The article addresses the problem that the overall machining quality of free-form surfaces is difficult to guarantee, this paper proposes a MATLAB-based free-form surface division method. It is proved that the method is practical and can effectively improve the machining efficiency and quality of complex surface. However, the paper is not innovative enough, and there are some problems that need to be improved. Thus, this paper should be major revised according to following comments:

1.     This paper lacks innovation. The NURBS method and fuzzy c-means algorithms mentioned in the paper are both widely used in surface modeling and image processing, respectively.

2.     It is argued in the article that this method has a significant improvement in machining efficiency, but is the machining quality of the surface components the same after applying this method?

3.     The transition area between convex and flat regions is smoother than the remaining two combinations, where is the evidence for it?

4.     In Figure 3, it is better to use two algorithms for free-form surfaces instead of Bunny point cloud data.

5.     Figure 12 does not clearly show the machining area of the different milling tools.

6.     Most of the references cited are about five-axis machine tools, which are more convincing when software simulation is used instead of three-axis NC milling.

7.     The following publications related to ultra-precision machining of optical surfaces may help to enrich the introduction of this paper:

[1] Zhang P, Li L, Yang Z, et al. Achieving sub-nanometer roughness on aspheric optical mold by non-contact polishing using damping-clothed tool[J]. Optics Express, 2022, 30(15): 28190-28206.

8.     The paper format should be enhanced.

Author Response

Dear Professor,

Thank you very much for your comments and suggestions.

Those comments are all valuable and very helpful for revising and improving our paper, as well as the important guiding significance to our researches. We have studied comments carefully and have made correction which we hope meet with approval. Revised portion are marked in blue in the paper. The main corrections in the paper and the responds to the reviewer’s comments are as flowing:

  1. Comment: This paper lacks innovation. The NURBS method and fuzzy c-means algorithms mentioned in the paper are both widely used in surface modeling and image processing, respectively.

Response: While most previous surface partitioning studies have used b-samples to define surfaces, which can only handle simpler free-form surfaces, this paper uses NURBS to define surfaces, which has wider applicability and can also handle more complex surfaces. Secondly, in the choice of clustering algorithm, previous authors either chose k-means algorithm or fuzzy c-means algorithm without further explanation of why the algorithm was chosen. In this paper, a cross-sectional comparison between k-means algorithm and fuzzy c-means algorithm is made, and the advantages of fuzzy c-means algorithm are demonstrated by visual clustering effect graphs. Finally, NURBS method is used to describe the surface, which can improve the universality of the algorithm. At the same time, fuzzy c-means clustering algorithm is used to deal with the surface, which ensures the rationality of segmentation.

  1. Comment: It is argued in the article that this method has a significant improvement in machining efficiency, but is the machining quality of the surface components the same after applying this method?

Response: This method not only improves machining efficiency compared with traditional machining methods, but also prevents overcutting of the tool and reserves machining allowance for grinding and polishing while ensuring the surface quality of finishing.

  1. Comment: The transition area between convex and flat regions is smoother than the remaining two combinations, where is the evidence for it?

Response: The original language expression was not strict and has been revised in the text.

  1. Comment: In Figure 3, it is better to use two algorithms for free-form surfaces instead of Bunny point cloud data.

Response: Since the fuzzy c-means algorithm effect is shown later in the program flow, the Bunny point cloud data is chosen to show the comparison effect in order to avoid repetition.

  1. Comment: Figure 12 does not clearly show the machining area of the different milling tools.

Response: The processing flow diagram and text presentation have been revised.

  1. Comment: Most of the references cited are about five-axis machine tools, which are more convincing when software simulation is used instead of three-axis NC milling.

Response: In the next step, a 3-axis CNC machining center will be used for experimental verification. Considering the consistency of subsequent experiments and simulations, a 3-axis machine is chosen for simulation verification.

  1. Comment: The following publications related to ultra-precision machining of optical surfaces may help to enrich the introduction of this paper:

Response: It has been carefully read and referenced, and is cited in the references.

  1. Comment: The paper format should be enhanced.

Response: Article formatting has been enhanced.
